# An Improved Future Land-Use Simulation Model with Dynamically Nested Ecological Spatial Constraints

**Chaoxu Luan, Renzhi Liu *, Jing Sun, Shangren Su and Zhenyao Shen**

State Key Laboratory of Water Environment Simulation, School of Environment, Beijing Normal University,
No. 19, Xinjiekouwai Street, Haidian District, Beijing 100875, China; 201931180018@mail.bnu.edu.cn (C.L.)
* Correspondence: liurenzhi@bnu.edu.cn; Tel.: +86-10-5880-0899

**Abstract:** A land-use simulation model oriented toward ecological constraints is effective for evaluating the ecological impact of urban spatial planning. However, few studies have incorporated dynamically nested ecological spatial constraints into the model or fully considered the urban development, agricultural production, and ecological function among the ecological spatial constraints. Therefore, this study developed an improved land-use simulation model with dynamically nested ecological spatial constraints (LSDNE). We fully considered the multilevel ecological spatial constraints from the perspectives of ecological (ecological protection red line, EPRL), production (capital farmland, CF), and living (urban development land-use suitability, UDLS). Five scenarios in terms of future land-use distribution in 2030 were set, namely, inertial development (S1), considering EPRL (S2), considering CF (S3), considering EPRL and CF (S4), and considering EPRL, CF, and UDLS (S5). This new approach was implemented in the rapidly developing provincial capital city of Changchun, China. The results show that, due to the occupation of arable land, Changchun had the largest increase in built-up land ($2019.75 \text{ km}^2$ to $3036.36 \text{ km}^2$) from 2010 to 2020. Terrain elevation was the most significant factor in all kinds of land expansion. According to future land spatial distribution results in 2030, under S4, Changchun's built-up land will be more compact compared with S1–S3 and S5, which showed more scattered built-up land. These predicted results show that Changchun's spatial planning put forward high requirements for the efficient use of land and constraints in red-line areas. Due to a clear evaluation of the impact of ecological spatial constraints on future land expansion, the LSDNE model provides more accurate support for the efficient use of land resources and future territorial spatial planning.

**Keywords:** land-use simulation; cellular automata; driving factors of land expansion; multilevel ecological spatial constraints; Changchun City

## 1. Introduction

Land resources play an irreplaceable role in all socioeconomic development and human activities. Since the industrial age, humans have intensified the rapid urban expansion and the tension of the interaction between humans and the environment through high-intensity land-use activities [1,2]. The lack of reasonable land-use policy and planning has incurred huge costs in terms of resource depletion, environmental pollution, and ecosystem degradation [3,4]. These resource, environmental, and ecological problems seriously affect sustainable development on a global scale [5,6]. Various countries have tried to specify relevant policies and measures to help enable healthy land-use planning [7,8]. For example, the Chinese government has clearly put forward the development concept of integration of multiple regulations, which has prompted more attention to research in this field. In the newly released territorial spatial planning policy in 2019, it was clearly proposed to "strengthen the guidance and constraints of territorial spatial planning on ecological and environmental protection". This series of territorial spatial planning policies filled in the missing part of the ecological and environmental evaluation in China's previous

urban planning content [4]. More importantly, in addition to natural and socioeconomic conditions, these changing plans and policies with the development of the region also directly affect local land-use distribution. Therefore, studying the spatiotemporal change of land use and clarifying the relationship between human activities and ecological protection can provide suggestions for relevant departments to formulate policies and promote sustainable development.

Land use is an important geospatial element, as well as an important link between nature and human beings [9]. In spatiotemporal simulations of land-use changes, future land-use trajectories are predicted through complex linkages and feedback structures [10,11]. A land-use dynamic simulation model is a reproducible and effective tool to simulate the future land expansion under the impact of natural, socioeconomic, and policy conditions, which is of great significance in research on ecological evaluation, land-use planning, and delimitation of urban growth boundary [12–14]. Cellular automata (CA) was the earliest model for future land-use simulation, which is based on the "bottom-up" framework and obtains global features through local rules [15,16]. It is easy to operate and able to simulate land-use change through the initial state of the grid, transition rules, and neighborhood effects [17].

Various CA-based spatial explicit discrete models contain different rule mining methods and land-use analysis strategies [18]. In the rule mining module, many studies use artificial intelligence (AI) algorithms to analyze the driving factors, which mainly include artificial neural networks (ANNs) [19,20], random forest (RF) [21], and support vector machines (SVMs) [22]. Furthermore, the widely used CA models mainly have two transformation rule mining strategies [23]. The first strategy is transition analysis strategy (TAS). TAS completes sample training according to the land-use conversion probability, such as logistic-CA [24] and ANN-CA [25]. In this strategy, the type of land-use conversion increases exponentially with the increase in categories, which leads to an increase in complexity and a decrease in flexibility. Another strategy is pattern analysis strategy (PAS), which mainly includes CLUE-s [26], Fore-SCE [27], and FLUS [28,29]. PAS only needs to extract land-use data for one period of time, which simplifies the computational complexity, but lacks the ability to mine the driving factors of land-use change. Nevertheless, TAS and PAS have been used to estimate the environmental variables impact on land expansion, providing many valuable simulation results [30–32]. However, these models have shortcomings in considering both the spatiotemporal evolution and the patch-level change of land use. Moreover, these models are insufficient to show the driving factors behind each type of land expansion and the strength of their impact intensity [33]. Another significant issue is the ecological constraints. A land-use simulation oriented toward ecological constraints is effective for eco-environmental management. However, there is no consistent method for how to integrate ecological spatial constraints into the model. In many current land-use simulation studies, ecological constraints are incorporated into the model as manual operation rules, combined with Boolean constraints or weighted linear-like models, which are not dynamically nested in the model [34,35]. The ecological constraints under static or manual operation will not change with the running of the model. This design mode simplifies the impact of ecological constraints. In addition, in the process of establishing ecological an index system of ecological constraints, urban development, agricultural production, and ecological functions are not comprehensively considered.

Here, we developed an improved land-use simulation model with dynamically nested ecological spatial constraints (LSDNE) and applied it to Changchun, China. The multilevel ecological spatial constraints were dynamically nested into the CA model. Unlike the previous CA modules, our model used multitype random patch seeds (MRPS) and analyzed the driving factors behind each type of land expansion and the strength of their impact intensity. More importantly, under the background of ecological civilization and spatial planning, we fully considered multilevel ecological spatial constraints from the ecological, production, and living perspectives. The ecological protection red line (EPRL), capital farmland (CF), and urban development land-use suitability (UDLS) were incorporated into

the LSDNE model as multilevel spatial constraints [36–38]. EPRL is an important institution in China's ecological conservation, which serves as the bottom line for ecological security. CF is designed to implement the strictest arable land protection system on the basis of the determined arable land that cannot be occupied according to planning. Meanwhile, UDLS analysis is aimed at determining the most suitable spatial pattern for future construction land use. Moreover, we computed the landscape metrics of Changchun. Land-use change at a regional scale inevitably causes alterations in the landscape patterns. Calculating landscape metrics offers a profound measure of land-use change. Analyzing the heterogeneity of land use throughout Changchun, alongside the dominance and fragmentation of diverse land types, is indispensable for obtaining an insightful comprehension of future land-use changes and the intuitive influence of ecological space constraints on spatially oriented future land utilization. As the case area of this study, Changchun is not only the ecological barrier in eastern China, but also the core area of China's food safety industrial belt. The previous free and extensive development pattern of heavy industry, which sacrificed the environment to achieve economic development, is no longer applicable. The development of Changchun must meet the ecological, production, and living requirements, which was also the principle of constructing the ecological spatial constraints in this study. Our study can promote the structural adjustment and layout optimization of the land resources in Changchun.

## 2. Materials and Methods

Figure 1 shows the flowchart of this study. In the LSDNE model, the RF algorithm, Markov model, CA model with MRPS mechanism, and multilevel ecological spatial constraints were integrated. After preprocessing, land-use data were entered into the Markov model to calculate the land-use transition probability matrix, and they were also used to extract land expansion maps. The MRPS-based CA module was developed to predict future land-use spatial distribution under multiple scenarios. The MRPS mechanism sets that overall growth probabilities (*OP*) are a product of the probability of occurrence (*P*), neighborhood effect (*Ω*), and adaptive driving coefficient (*D*). The overall probability was adjusted as a function of the probability of occurrence and dynamically nested multilevel ecological spatial constraints. We set up five scenarios according to the impact of the EPRL, CF, and UDLS.

### 2.1. Study Area

As a rapidly developing provincial capital city, Changchun (43°05′–45°15′N, 124°18′–127°05′E) is located in the central part of northeastern China, which consists of four counties/cities and seven districts (Figure 2). In addition, Changchun belongs to the transitional zone from the eastern hills to the western platform with a large plain area, and the slopes are mostly 2°–15°. In terms of land-use types, Changchun has a lot of high-quality arable land. Under the context of the Northeast Revitalization Strategy, Changchun's economy has developed rapidly in recent years. Driven by development zones at all levels and major transportation corridors, urban expansion in Changchun with a marginal expansion pattern has been accelerating. At present, land is still the main factor in the development of Changchun. In the latest phase of the Northeast Revitalization Strategy and the Harbin–Changchun City Cluster plan, new requirements have been put forward for the development of Changchun. Therefore, Changchun has faced huge demand for land expansion and development potential.

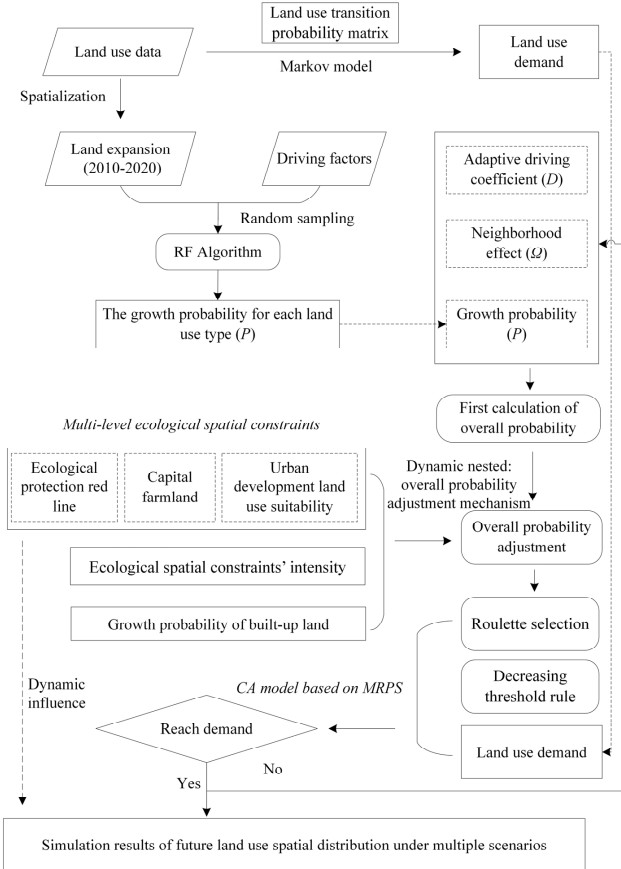

**Figure 1.** The flowchart of this study.

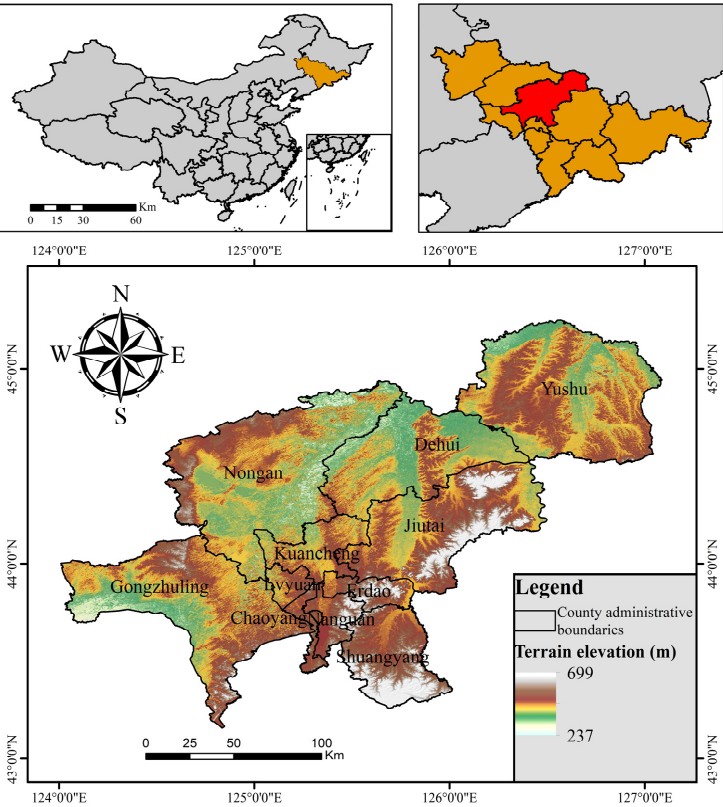

**Figure 2.** Study area location.

*2.2. LSDNE Model for Future Land-Use Simulation*

2.2.1. Quantitative Simulation Using Markov Model

The Markov model is a raster scale-based model with strong quantitative prediction ability, which has stability and no aftereffect. Stability means that the change process tends to be stable, and no aftereffect means that the situation at a certain moment is not affected by the past and the future, being only related to the current situation [39,40]. The application of the Markov model in land-use simulation is mainly to predict the number of grids of each land-use type, so as to make up for the deficiency of the ordinary spatial model in quantitative prediction [41]. The formulas are as follows:

$$S_t = S_{t+1} \times P_{ij}, \tag{1}$$

$$P_{ij} = \begin{bmatrix} P_{11} & \cdots & P_{1n} \\ \vdots & \ddots & \vdots \\ P_{n1} & \cdots & P_{mn} \end{bmatrix}, \tag{2}$$

$$P_{ij} \in [0,1) \ and \ \sum_{k=1}^{n} P_{ij} = 1 (i, j = 1, 2, \ldots, n), \tag{3}$$

where $S_t$ and $S_{t+1}$ represent the state of a certain type of land use at $t$ and $t + 1$, respectively, $P_{ij}$ represents the transition probability from land-use type $i$ to $j$, and $n$ is the number of land-use types.

2.2.2. Driving Factor Analysis Using RF Algorithm

The LSDNE model combines the advantages of TAS and PAS, considers the patches whose land-use types have changed, ignores their sources, and simplifies the analysis process of land expansion. When exploring the relationship between the land expansion of all types and their driving factors, the label of the expanded sample was set to 1, and the label of the sample without expansion was set to 0. We extracted the value of each driving factors for each sample grid. After establishing the training dataset, we obtained the growth probability of all kinds of land use and the contribution degree during this period using the random forest (RF) algorithm, which is composed of multiple decision trees [33]. The formula for the final growth probability $P_{i,k}^d$ for grid $i$ is as follows:

$$P_{i,k}^d(x) = \frac{\sum_{n=1}^{M} I(h_n(x) = d)}{M}, \tag{4}$$

where $M$ is the number of decision trees, the value of $d$ is 0 or 1 ($d = 1$ indicates that there are other land use types have been converted to land use type $k$; $d = 0$ represents other conversions), $x$ is a vector made up of all the driving factors of land expansion, $h_n(x)$ is the prediction type of the $n$-th decision tree of vector $x$, and $I$ is the indicative function.

When using the LSDNE model for simulation, the selection results of the driving factors of land expansion directly affects the accuracy of simulation. According to data availability, quantification, and spatial differences, we selected 12 driving factors of land expansion, which are shown in Table 1. Furthermore, according to the evolution characteristics of land use, it was determined whether or not to remove specific driving factors from the final indicator system.

**Table 1.** Driving factors of land expansion.

| Category | Indicators | Explanation |
|---|---|---|
| Topographic and geologic data | Terrain elevation | The terrain elevation of the location, obtained directly from the DEM data |
| | Slope | The ratio of the vertical height of the slope to the distance in the horizontal direction, calculated from the terrain elevation data |
| | Population | Population per unit area |
| | GDP | Used to evaluate the economic status of a region, reflecting the ability of economic development |
| Socioeconomic data | Proximity to highway Proximity to railway Proximity to national road Proximity to provincial road Proximity to urban area | The distance from the location to the nearest railway, road or urban area |
| | Soil type | The basic factor of land-use distribution, related to the production capacity and the availability of the land |
| Environmental and climate data | Annual mean temperature Annual precipitation | Climatic indicators that affect human production and life, generated by calculation and spatial interpolation |

### 2.2.3. Spatiotemporal Simulation Using MRPS-Based CA Model

(1) First calculation of overall probability

The CA model adopts a threshold drop-based MRPS mechanism. According to the MRPS mechanism, the overall growth probabilities (*OP*) are a product of the probability of occurrence (*P*), neighborhood effect ($\Omega$), and adaptive driving coefficient (*D*). The overall probability $OP_{i,k}^{d=1,t}$ is calculated as follows:

$$OP_{i,k}^{d=1,t} = P_{i,k}^{d=1} \times \Omega_{i,k}^t \times D_k^t, \tag{5}$$

where $P_{i,k}^{d=1}$ is the growth probability of land-use type *k* of grid *I*, $D_k^t$ is an adaptive driving coefficient, and $\Omega_{i,k}^t$ indicates the neighborhood effect.

(2) Neighborhood effect

The neighborhood effect $\Omega_{i,k}^t$ of grid *i* represents the coverage of land-use type *k* in a certain neighborhood, which is calculated as follows:

$$\Omega_{i,k}^t = \frac{con\left(c_i^{t-1} = k\right)}{n \times n - 1} \times w_k, \tag{6}$$

where $con\left(c_i^{t-1} = k\right)$ is the total number of grids of land-use type *k* in the "$n \times n$" window, and *w* is the weight that can be changed by the user.

(3) Adaptive driving coefficient

The adaptive driving coefficient $D_k^t$ is an indicator of the inheritance of land-use types. It plays a crucial role in correcting the trajectory of land use in cases where the development trend of a specific land use type contradicts the macro demand. This correction is achieved by dynamically increasing $D_k^t$ to enhance its inheritance in the next iteration [10]. The adaptive method of calculating $D_k^t$ is as follows:

$$D_k^t = \begin{cases} D_k^{t-1} \; if \; \left|G_k^{t-1}\right| \leq \left|G_k^{t-2}\right| \\ D_k^{t-1} \times \frac{G_k^{t-2}}{G_k^{t-1}} \; if \; 0 > G_k^{t-2} > G_k^{t-1} \\ D_k^{t-1} \times \frac{G_k^{t-1}}{G_k^{t-2}} \; if \; G_k^{t-1} > G_k^{t-2} > 0 \end{cases}, \tag{7}$$

where $G_k^{t-1}$ and $G_k^{t-2}$ are the difference between future and present demand of land-use type *k*. To avoid any alteration of *OP*, it is advisable to set the initial value of $D_k^t$ as 1.

(4) Overall probability adjustment by dynamically nested ecological spatial constraints

The LSDNE model considers the ecological spatial constraints. This study focused on the ecological spatial constraints from a sustainable development perspective (ecological, production, and living). This module will solve the problem of previous studies simply considering the restrictive effect of the policy rather than the driving effect of the policy at different levels. If the growth probability *(P)* of a land-use type is greater than a random value within the range of 0 to 1, then a random seed is planted in the cell [33]. During the simulation process of the model, after completing the calculation of *OP*, the newly added ecological spatial constraints maps are scanned. After scanning the ecological spatial constraints, the random seeds planted in the cells adjust *OP* as follows:

$$OP_k \ (adjusted) = \begin{cases} (R + OP_k) \times w \ if \ R + OP_k \leq 1 \\ I \times w \ if \ R + OP_k > 1 \end{cases}, \tag{8}$$

where *I* is the ecological spatial constraints' intensity, *w* is the ecological spatial constraints' weight, and *R* is a random number. In this study, we took into account the UDLS, CF, and EPRL from a sustainable development perspective. We mainly considered the impact of ecological spatial constraints on urban expansion; thus, *k* was set as built-up land. On the basis of our previous research results about UDLS evaluation, UDLS was classified into five levels: highly suitable, suitable, moderately suitable, marginally suitable, and not suitable [36,37]. Since EPRL and CF are divided into two levels, and their grading standards are consistent with the "highly suitable and not suitable" in UDLS, the policy impact intensity value obtained by combining EPRL, CF, and UDLS was also classified into five groups, similar to UDLS. Therefore, the maximum value of *I* was 4, and we set the weight to 1/4 so that the maximum value of *w* multiplied by *I* was 1. Figure A2 shows the ecological spatial constraints maps. After this process, the adjusted *OP* is then used for subsequent simulations and iterations based on the roulette selection and the descending threshold rule.

(5) Descending threshold rule

The model also includes a descending threshold rule for generating patch seeds in order to gradually restrict the patch growth. Grids with higher total probability are most likely to change first according to this decreasing threshold rule:

$$If \ \sum_{k=1}^{N} \left| G_c^{t-1} \right| - \sum_{k=1}^{N} \left| G_c^{t} \right| < Step \ Then, l = l + 1, \tag{9}$$

$$\begin{cases} Change \ P_{i,c}^{d=1} > \tau \ and \ TM_{k,c} = 1 \\ No \ change \ P_{i,c}^{d=1} \leq \tau \ or \ TM_{k,c} = 0 \end{cases} \tau = \delta^l \times r1, \tag{10}$$

where *Step* is the step size, and $\delta$ is the attenuation factor, ranging from 0 to 1. A higher $\delta$ implies more conservative conversion strategies. To enable possible changes for cells with lower probability of conversion while taking into account the default value of the model, we set $\delta$ to 0.5. The probability distribution of random variables in the model should be described using a normal distribution to ensure that the distribution is centered around the mean; hence, *r1* is a normally distributed random value with a mean of 1 and a range from 0 to 2, *l* is the number of decay steps, and $TM_{k,c}$ is a transition matrix used to define whether land-use type *k* can be converted to land-use type *c*.

(6) Scenario setting

To better assess the impacts of different ecological spatial constraint policies in territorial spatial planning on Changchun's land-use pattern by 2030, five scenarios (S1–S5) were designed in this study. S1 was an inertial development scenario that reflected historical trends in land use without considering the impact of ecological spatial constraints on future land use. S2 and S3 were "single ecological spatial constraint policy" scenarios that focused on EPRL and CF, respectively. S2 increased the consideration of the EPRL policy in ecological spatial constraints. Similarly, S3 only considered the CF ecological spatial constraint policy. S4 was a "dual ecological spatial constraints policy" scenario that simultaneously

considered both EPRL and CF protections. Lastly, S5 was a comprehensive scenario that integrated all three constraints (EPRL, CF, and UDLS).

### 2.2.4. Model Validation

By inputting the parameters in 2010 required by the LSDNE model, we obtained the predicted results of Changchun in 2020. Then, we compared the actual situation with the predicted results. To evaluate the credibility of the model, we calculated both the kappa coefficient and overall accuracy. We utilized the kappa index to evaluate the accuracy of the simulation results by comparing them with actual data using the following formula:

$$Kappa = (P_0 - P_c)/(P_p - P_c),$$ (11)

where $P_0$ refers to the overall classification accuracy, while $P_c$ and $P_p$ denote the actual simulation accuracy and ideal simulation accuracy, respectively. Typically, a kappa value greater than 0.75 indicates high agreement between the actual and simulated degrees, while values ranging from 0.4 to 0.75 indicate general high agreement. Values below 0.4, on the other hand, represent poor agreement. In our study, the kappa coefficient was 0.9186, and the overall accuracy was 0.9524. This shows that the LSDNE model had high credibility and could be applied to simulate land-use distribution in 2030.

### 2.3. Other Methods for Analyzing Land-Use Change Characteristics

### 2.3.1. Calculation of Land-Use Change Rate

There are many methods used to quantitatively describe the sources and trends of land-use conversion. According to studies in the relevant literature, we calculated the transition and increase rate of all land-use types [42,43]. The transition rate ($TRL_i$) and the increase rate ($IRL_i$) were calculated as follows:

$$TRL_i = \frac{LA(i, t_1) - ULA_i}{LA(i, t_1)} \times \frac{1}{t_2 - t_1} \times 100\%,$$ (12)

$$IRL_i = \frac{LA(i, t_2) - ULA_i}{LA(i, t_1)} \times \frac{1}{t_2 - t_1} \times 100\%,$$ (13)

where $LA(i, t_1)$ is the area of land-use type $i$ at $t_1$, $LA(i, t_2)$ is the area of land-use type $i$ at $t_2$, and $ULA_i$ is the unchanged area of land-use type $i$.

Thus, the change rate of land use type $i$ ($CRL_i$) is

$$CRL_i = IRL_i - TRL_i,$$ (14)

where a positive $CRL_i$ indicates that the land of type $i$ has increased, while a negative value indicates that it has decreased.

### 2.3.2. Landscape Metrics Calculation

Human activities lead to land-use change, and they also inevitably lead to landscape pattern change. Landscape metrics are quantitative tools to define landscape structure and spatial pattern. Analyzing landscape pattern is very important to understand the situation of urban development. There are various landscape metrics, and the indicators are interrelated and highly correlated. Landscape metrics can be divided into landscape-level, class-level, and patch-level, which reflect the overall structural characteristics, the characteristics of the same patch type, and the structural characteristics of a patch. FRAGSTATS is an effective tool for analyzing and quantifying landscape structure, and it can calculate a large number of landscape metrics. According to the relevant literature, eight indicators were selected and they were calculated using the FRAGSTATS 4.2 software package [43,44]. The specific descriptions and formulas of landscape metrics are presented in Table 2.

**Table 2.** Specific descriptions and formulas of landscape metrics.

| Indicators | Formula |
|---|---|
| Shannon's diversity index (SHDI) | $\text{SHDI} = -\sum\limits_{i=1}^{m}(P_i \ln P_i)$ |
| Shannon's evenness index (SHEI) | $\text{SHEI} = \dfrac{-\sum_{i=1}^{m}(P_i \ln P_i)}{\ln m}$ |
| Contagion (CONTAG) | $\text{CONTAG} = \left[1 + \dfrac{\sum_{i=1}^{m}\sum_{k=1}^{m}\left[P_i \frac{g_{ik}}{\sum_{k=1}^{m}g_{ik}}\right]\left[\ln\left(P_i \frac{g_{ik}}{\sum_{k=1}^{m}g_{ik}}\right)\right]}{2\ln(m)}\right](100)$ |
| Number of patches (NP) | $\text{NP} = n$ |
| Patch density (PD) | $\text{PD} = \dfrac{N}{A}$ |
| Aggregation index (AI) | $\text{AI} = \left[\dfrac{g_{ij}}{max\, g_{ij}}\right] \times 100$ |
| Largest patch index (LPI) | $\text{LPI} = \dfrac{max_{j=1}^{n}(a_{ij})}{A}(100)$ |
| Splitting index (SPLIT) | $\text{SPLIT} = \dfrac{A^2}{\sum_{j=1}^{n}a_{ij}^2}$ |

### 2.4. Data Sources

Land-use data were derived from the basic geographic information center of China (http://www.globallandcover.com (accessed on 18 December 2022)). According to the classification standard of data sources, there were seven land-use types in Changchun: arable land, built-up land, grassland, woodland, wetland, open water, and unused land. GDP, population density, soil type, precipitation, and temperature data were obtained from the resources and environment data center of the Chinese Academy of Sciences (http://www.resdc.cn (accessed on 18 December 2022)). Terrain elevation and slope data were derived from the geospatial data cloud (http://www.gscloud.cn (accessed on 18 December 2022)). They were processed with a 30 m × 30 m digital elevation model (DEM). Road data were obtained from the OpenStreetMap (https://www.openstreetmap.org (accessed on 18 December 2022)). The protection area of EPRL, CF, and UDLS data were derived from the maps of ecological spatial constraints in the territory spatial planning of Changchun. After being preprocessed, all data were uniformly converted into raster data with a spatial resolution of 30 m × 30 m and a range of 8057 × 7897 grids.

## 3. Results

### 3.1. Land-Use Evolution Characteristics

3.1.1. Land-Use Transition and Change Rate

Table 3 shows the land-use transition matrix of Changchun, and the results of the change characteristics of all kinds of land use are presented in Table 4. Arable land was the most extensive, but decreased the most (1179.91 km$^2$), accounting for 5.71%. Built-up land increased the most, from 2019.75 km$^2$ to 3036.36 km$^2$. Its increase and transition rate were 73.97% and 23.64%, respectively. Wetland had the highest rate of increase (647.70%) and transition rate (79.08%), increasing by 201.15 km$^2$. The area of woodland decreased by 1984.96 km$^2$, while that of grassland decreased by 1358.41 km$^2$.

**Table 3.** Land-use transition matrix from 2010 to 2020 (unit: km$^2$).

| 2010 \ 2020 | Arable Land | Woodland | Grassland | Wetland | Open Water | Built-Up Land | Unused Land | Total |
|---|---|---|---|---|---|---|---|---|
| Arable land | 18,654.06 | 127.99 | 159.55 | 200.66 | 88.75 | 1444.19 | 0.26 | 20,675.47 |
| Woodland | 121.86 | 578.88 | 116.81 | 1.48 | 3.42 | 13.57 | 0.16 | 836.19 |
| Grassland | 223.66 | 104.36 | 397.57 | 5.74 | 21.76 | 32.15 | 3.53 | 788.76 |
| Wetland | 2.77 | 0.07 | 16.83 | 7.40 | 8.13 | 0.18 | 0.01 | 35.38 |
| Open water | 27.94 | 1.65 | 19.00 | 20.07 | 278.20 | 3.75 | 0.01 | 350.61 |
| Built-up land | 465.06 | 4.57 | 4.62 | 1.16 | 2.00 | 1542.33 | 0.02 | 2019.75 |
| Unused land | 0.20 | 0.29 | 3.68 | 0.02 | 0.18 | 0.18 | 2.26 | 6.81 |
| Total | 19,495.56 | 817.81 | 718.04 | 236.53 | 402.42 | 3036.36 | 6.25 | 24,712.96 |

**Table 4.** Results of the change characteristics of all kinds of land use.

| Land Use Type | Arable Land | Woodland | Grassland | Wetland | Open Water | Built-Up Land | Unused Land |
|---|---|---|---|---|---|---|---|
| Converted to other types (km$^2$) | 2021.41 | 257.30 | 391.19 | 27.98 | 72.42 | 477.42 | 4.55 |
| Newly generated (km$^2$) | 841.50 | 238.92 | 320.48 | 229.13 | 124.23 | 1494.03 | 3.99 |
| Area change (km$^2$) | −1179.91 | −18.38 | −70.72 | 201.15 | 51.81 | 1016.61 | −0.56 |
| Increasing rate ($IRL_i$) | 4.07% | 28.57% | 40.63% | 647.70% | 35.43% | 73.97% | 58.57% |
| Transition rate ($TRL_i$) | 9.78% | 30.77% | 49.60% | 79.08% | 20.65% | 23.64% | 66.80% |
| Change rate ($CRL_i$) | −5.71% | −2.20% | −8.97% | 568.61% | 14.78% | 50.33% | −8.24% |
| Converted to other types (km$^2$) | 2021.41 | 257.30 | 391.19 | 27.98 | 72.42 | 477.42 | 4.55 |

Figure 3 shows the land-use distribution maps of Changchun between 2010 and 2020. In order to highlight the main characteristics during the period, we added land-use transition maps and a Sankey diagram. The newly created arable land was scattered across Changchun, whereas the converted woodland was mainly in the southeastern parts of Changchun. The converted built-up land was mainly in the southern–central part of Changchun. The converted grassland and wetland were located in the western and northern parts of Changchun, and the other types of converted land accounted for lower proportions.

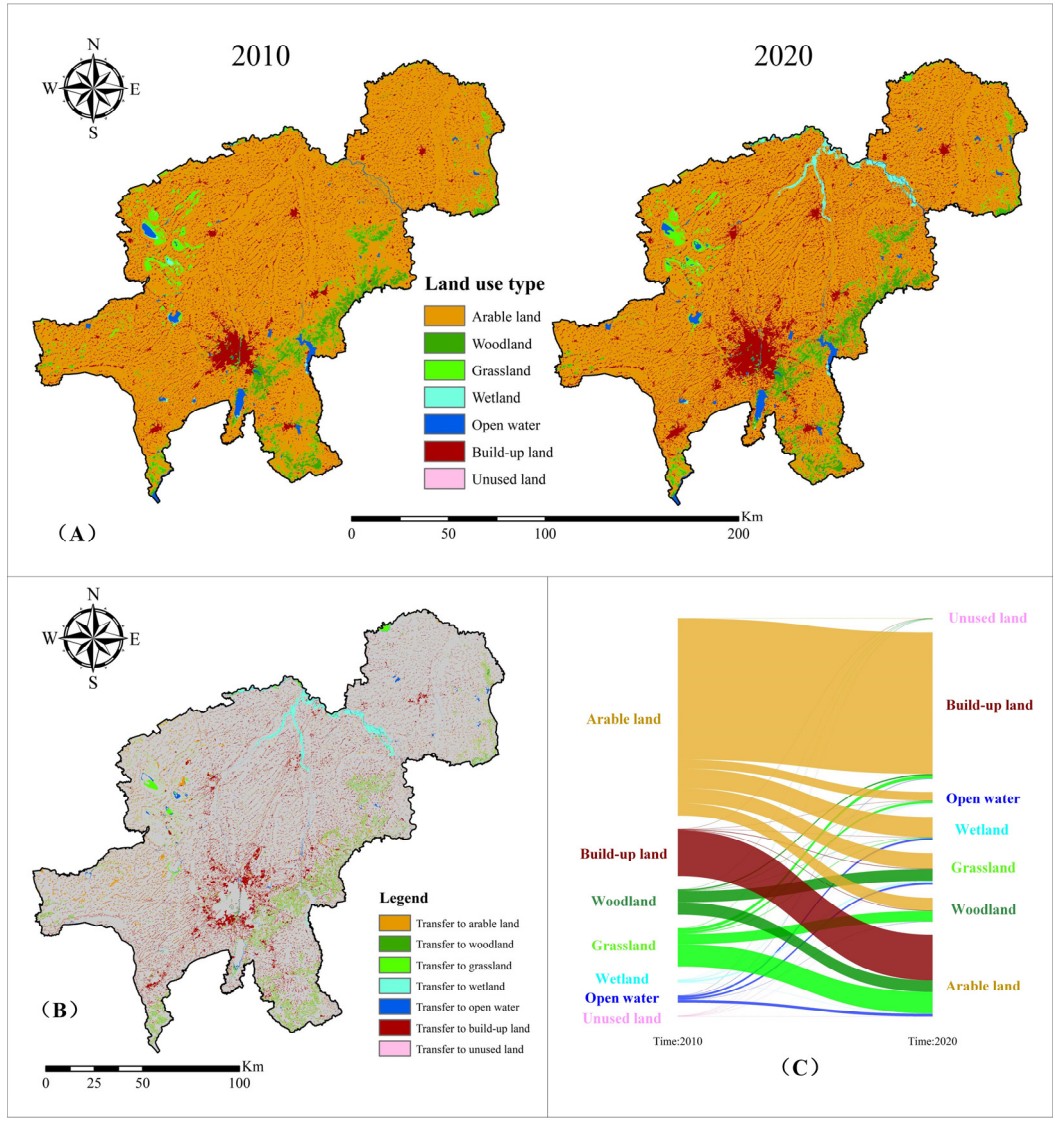

**Figure 3.** Land-use maps. (**A**) Land-use maps of 2010 and 2020. (**B**) Land-use transition maps between 2010 and 2020. (**C**) Sankey diagram of land-use transition between 2010 and 2020.

3.1.2. Landscape Pattern

The landscape diversity can reflect whether the spatial and temporal distributions of the landscapes are balanced and complex, i.e., the heterogeneity of the landscapes. The landscape-level metrics of Changchun in 2010 and 2020 are shown in Table 5. The SHDI value was higher in 2020 than in 2010, indicating that the landscape diversity in Changchun increased in the 10 year study period. The SHEI value also increased, indicating that the patches of each landscape type became more and more uniform. The CONTAG value decreased from 2010 to 2020, indicating that the connectivity between the patches became worse, and the area of the patches became smaller and more fragmented. Overall, from 2010 to 2020, the patches in Changchun became more scattered, and the dominant patches became less obvious.

**Table 5.** Landscape-level metrics in 2010 and 2020.

| Year Landscape Indicators | SHDI | SHEI | CONTAG |
|---|---|---|---|
| 2010 | 0.6504 | 0.3343 | 79.5058 |
| 2020 | 0.7739 | 0.3977 | 75.5789 |

On the landscape class scale, five indicators (NP, PD, AI, LPI, and SPLIT) were used to analyze the class-level landscape changes of Changchun. It is shown from Table 6 that, during 2010–2020, the values of NP and PD for grassland were the largest. The patches of grassland were the most fragmented among all of the types. Wetland had the smallest NP and PD values. The NP and PD of arable land and built-up land increased significantly. The LPI value of built-up land was the largest in 2010 and increased significantly in 2020. It was found that built-up land became increasingly dominant in the various landscape types in Changchun. The AI values of all kinds of land use did not change greatly from 2010 to 2020. In addition, the SPLIT values of arable land and built-up land were small and decreased sharply in 2020. In summary, during the 10 year study period, the urban agricultural production activities and residents' lives became more concentrated.

**Table 6.** Class-level landscape metrics in 2010 and 2020.

| Landscape Indicators | Year | Arable Land | Woodland | Grassland | Wetland | Open Water | Built-Up Land | Unused Land |
|---|---|---|---|---|---|---|---|---|
| NP | 2010 | 909 | 8955 | 20,793 | 35 | 3129 | 9086 | 906 |
| | 2020 | 1697 | 8455 | 20,291 | 286 | 2175 | 12,104 | 832 |
| PD | 2010 | 0.0367 | 0.362 | 0.8404 | 0.0014 | 0.1265 | 0.3673 | 0.0366 |
| | 2020 | 0.0686 | 0.3417 | 0.8202 | 0.0116 | 0.0879 | 0.4892 | 0.0336 |
| LPI | 2010 | 65.3646 | 0.2899 | 0.2843 | 0.0327 | 0.2131 | 0.9196 | 0.002 |
| | 2020 | 77.7899 | 0.2842 | 0.3543 | 0.3369 | 0.2267 | 2.7343 | 0.002 |
| AI | 2010 | 98.4647 | 87.6912 | 81.9546 | 94.4868 | 92.3743 | 90.0893 | 62.1687 |
| | 2020 | 97.9674 | 87.3472 | 81.2893 | 94.632 | 91.7074 | 90.5103 | 61.3307 |
| SPLIT | 2010 | 2.1897 | 33,442.6383 | 80,343.1692 | 3,694,231.847 | 68,212.7968 | 8291.0848 | 580,493,477.7 |
| | 2020 | 1.6525 | 39,683.4147 | 60,864.9965 | 72,208.3061 | 72,726.3506 | 1307.0577 | 707,709,449.4 |

*3.2. Driving Factors Analysis of Land Expansion*

Twelve driving factors were used to simulate future land-use distribution by the LSDNE model, and they were analyzed using the RF algorithm. Figure A1 shows the maps of driving factors. As Changchun included four main types of land use (built-up land, arable land, built-up land, grassland, and woodland), we mainly analyzed the driving factors of these four types of land expansion. The contribution degree of driving factors and the growth probability of four main kinds of land use are shown in Figure 4. For all four types of land expansion, terrain elevation was the most significant factor. In terms of arable land, the growth probability was higher in the western edge of Changchun. In addition to the terrain elevation, proximity to highways and the annual precipitation were also of great

contribution to the arable land expansion. This indicates that the natural conditions, the growing conditions of the crops, and the human activities simultaneously affected arable land expansion. For built-up land, in addition to the terrain elevation, human and economic factors such as population and proximity to highways and railways were also of great contribution. Moreover, the expansion of woodland and grassland was mainly influenced by the annual precipitation and population. Woodland expansion mainly occurred in the southeastern part of Changchun, while grassland expansion mainly occurred in the western part of Changchun. The contribution of the population suggests that some of the woodland and grassland may have been managed by humans for the purpose of ecological protection.

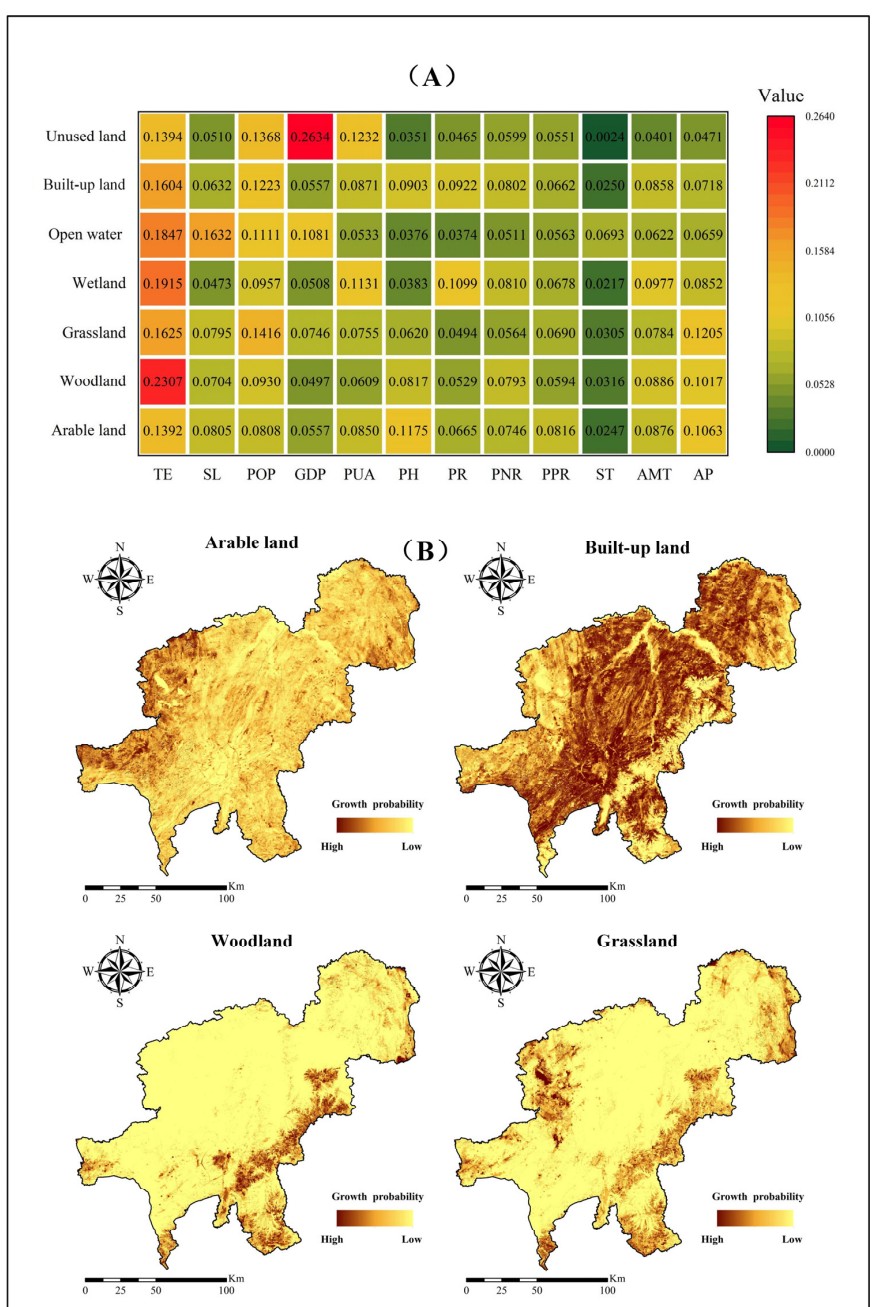

**Figure 4.** Maps of driving factor analysis. (**A**) The contribution degree of driving factors for each type of land expansion. (**B**) The growth probability of four main kinds of land use.

### 3.3. Land-Use Distribution under Five Scenarios

Table 7 presents the quantitative simulation results of land use from 2010 to 2030. By 2030, the arable land in Changchun would decrease by 833.33 km$^2$, while the build-up land area would increase by 692.24 km$^2$. These results indicate that the level of urbanization in Changchun is expected to continue to rise. Furthermore, the forest area would have a slight decrease of 26.48 km$^2$ in 2030, while grassland, wetland, and open water would increase by 53.2 km$^2$, 33.62 km$^2$, and 81.21 km$^2$, respectively. These changes reflect the gradual expansion of ecological space in Changchun.

**Table 7.** Quantitative simulation results of land use from 2010 to 2030.

|  | Arable Land | Woodland | Grassland | Wetland | Open Water | Built-Up Land | Unused Land |
|---|---|---|---|---|---|---|---|
| 2010 | 20,675.47 | 836.19 | 788.76 | 35.38 | 350.61 | 2019.75 | 6.81 |
| 2020 | 19,495.56 | 817.81 | 718.04 | 236.53 | 402.42 | 3036.36 | 6.25 |
| 2030 | 18,662.23 | 791.33 | 771.24 | 270.15 | 483.63 | 3728.60 | 5.79 |

Land-use distribution was predicted by the LSDNE model after calculating the quantitative results. The ecological spatial constraints affected the total probability of the built-up land. Figure 5 shows the land-use maps and diversity maps for Changchun in 2030 under different scenarios, and Figure A3 shows the land expansion maps. The spatial differentiation between S1 and S2 was relatively insignificant. While there was some variation between S1 and S2, it was largely confined to specific regions within Dehui district where it met the adjoining districts of Nongan and Yushu. In contrast to S2, the disparity between S3 and S1 was greater and more scattered across several regions. S4 exhibited the highest degree of similarity to S3, and this similarity level was comparable to the disparity observed between S4 and S1. In addition, S5 exhibited marked differences compared to S1, with multiple areas of substantial disparity that were primarily concentrated in the central region of Nanguan district and several parts of Dehui, Jiutai, and Shuangyang districts.

In order to compare and analyze the different simulation results under the ecological spatial constraints more intuitively, we calculated the landscape metrics of all land-use types under S1–S5. Figure 6 shows the results of the landscape metrics under S1–S5. The land expansion characteristics under S1 and S2 were generally similar. Compared with the other scenarios, S1 and S2 had smaller CONTAG values, but their SHEI and SDEI values were larger. This indicates that patches of various land-use types were highly fragmented and evenly distributed under S1 and S2. In contrast, there were obvious dominant patches under S3, S4, and S5. The patches had high connectivity and the spatial distributions of the patches became more uneven. In terms of the class-specific landscape metrics, for arable land, NP and PD were significantly larger under S1 and S2 than under other scenarios. It can be concluded that the patches of arable land were more fragmented under S1 and S2 than under the other scenarios. Under S3 and S4, built-up land had larger LPI values and smaller SPLIT values. Under S3 and S4, the fragmentation degree of built-up land was smaller, the patches were more compact, and the landscape dominance was more obvious. As can be seen from Figure A3, under S3 and S4, the built-up land expansion was mainly concentrated in the center of Changchun, such as the Chaoyang, Nanguan, Erdao, Lvyuan, and Kuancheng District. For woodland, under S1–S5, the LPI value gradually decreased, while the SPLIT value gradually increased. It can be concluded that, under S1–S5, the landscape dominance of woodland gradually decreased, and the patches became more and more fragmented. In contrast, there was little difference in the landscape metric values for grassland under the different scenarios.

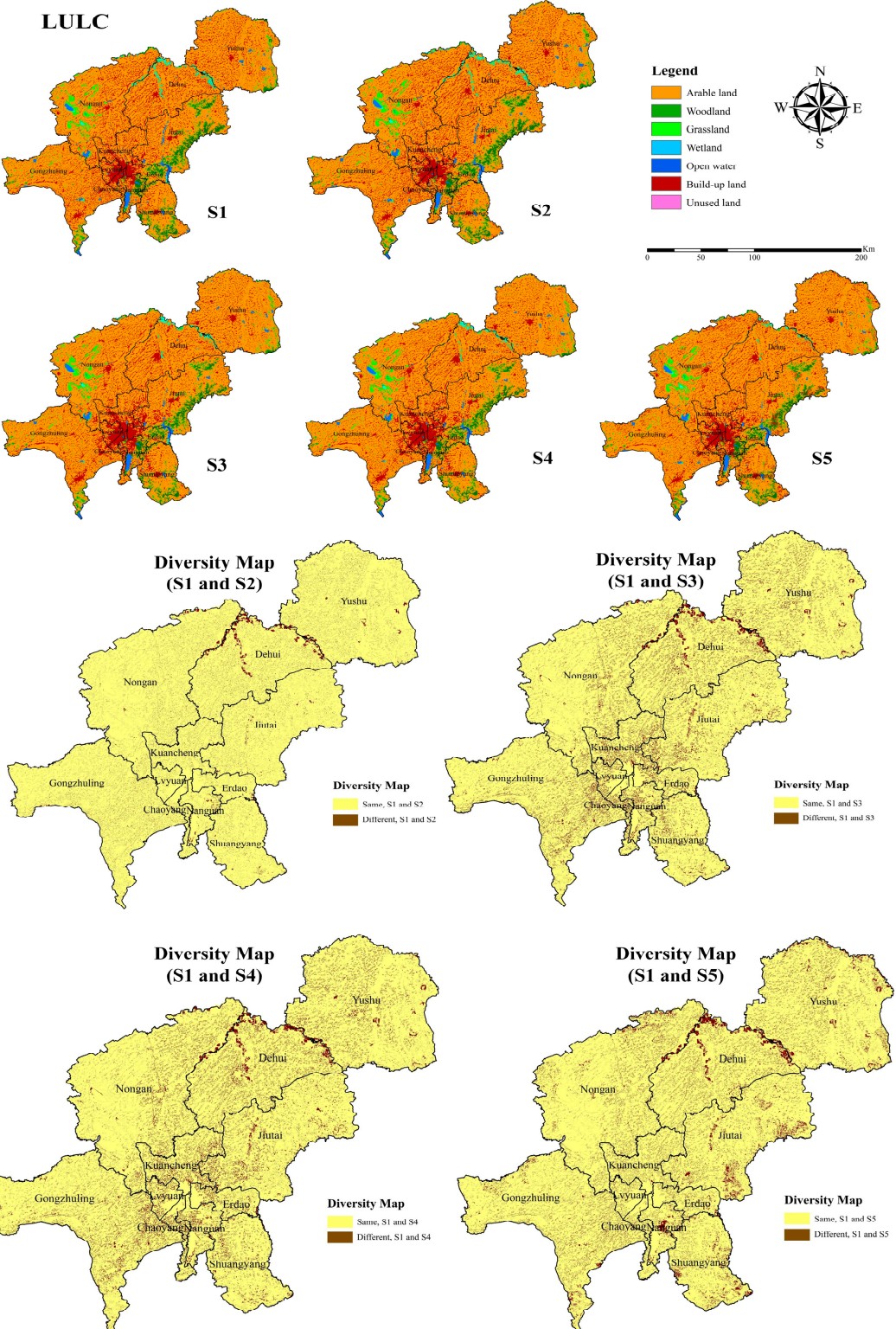

**Figure 5.** Land-use maps and diversity maps under S1–S5 in 2030.

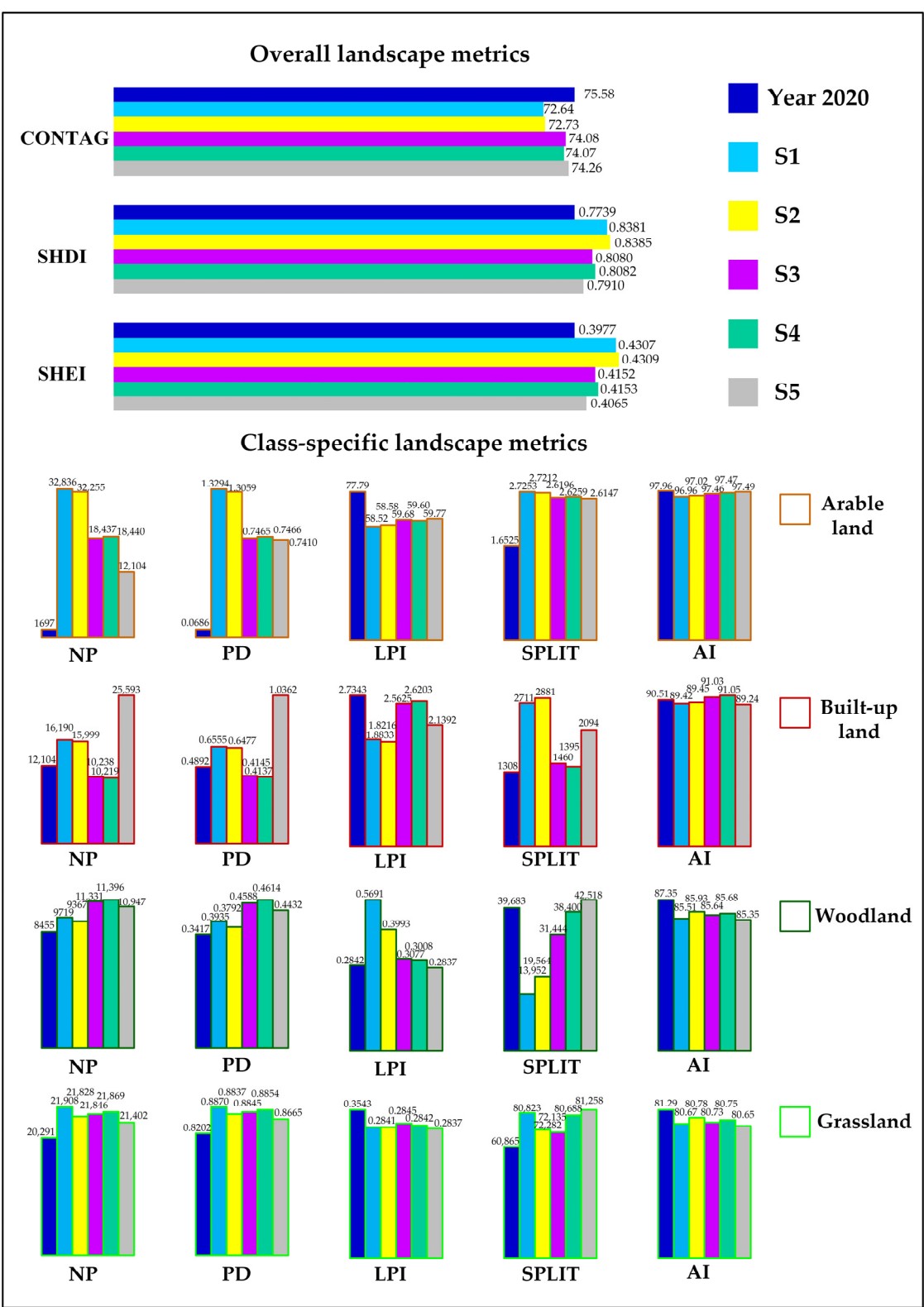

**Figure 6.** Results of overall and class-specific landscape metrics under S1–S5.

## 4. Discussion

### 4.1. An Improved Model for Future Land-Use Simulation

So far, many studies have simulated future land-use distribution and explored the driving factors of land expansion. Unlike other studies, we fully considered the impacts

of ecological spatial constraints on future land-use distribution from the ecological, production, and living perspectives, and we innovatively constructed the multilevel spatial constraints of UDLS, CF, and EPRL. We coupled multilevel ecological spatial constraints as an adjustment module of the LSDNE model for overall probability instead of simple operations such as Boolean constraints. In our study, the ecological spatial constraint maps did not directly constrain the simulation results of land use, but were instead nested in the model and dynamically changed with model progression, thus rendering the land-use simulation under multilevel ecological constraints more scientific and precise, as well as expanding the application range of the model. This can help decision makers better understand how the policy with ecological spatial constraints will affect the future land expansion. Another advantage of this method is that the LSDNE model can effectively show the driving factors behind each type of land expansion and the strength of their impact intensity when we need to analyze each type of land expansion. The verification results of the model accuracy showed that the LSDNE model has high practicability, and that our study is an improvement and development of CA-based land-use simulation. In summary, from the perspective of ecological priority, our model enhances the control ability of ecological spatial constraints to better protect regional ecological functions. The design of ecological spatial constraints model has broad application prospects in land use model simulation, especially in urban planning.

### 4.2. Simulation Results and Suggestions of the Study Area

Figure 4 shows how each driving factor affected each land expansion, representing more specific results of driving factor analysis compared with previous studies. Terrain elevation was the most significant factor in all kinds of land expansion. This indicates that the topographical condition plays a more significant role in restricting the evolution of land use, both for urban–artificial landscapes and for natural landscapes. In addition, population was another important driving factor in all kinds of land expansion. The impact of population on built-up land and unused land is mainly reflected in the intensification of urbanization caused by human activities, while the impact of population on arable land, woodland, and grassland is more likely reflected in the implementation of territorial spatial policies such as "returning farmland to grassland, returning farmland to forest, protecting capital farmland, and limiting urban expansion". In general, human activities and policy factors have played a crucial role in the land-use evolution of Changchun. Therefore, it is of great practical significance to analyze the impact of ecological spatial constraints on future land-use distribution.

We conducted a comparative analysis of future land-use spatial distribution with or without the ecological spatial constraints, so as to more clearly evaluate the impact of Changchun's territorial spatial policies on future land-use distribution. As EPRL and CF protection is a strictly restrictive policy, under the dual protection scenario (S4), Changchun's built-up land would show a trend of outward expansion in 2030, and the urban area would be more compact. However, if the impact of UDLS is also considered (S5), the built-up land in Changchun would be more scattered in 2030. This shows that the spatial constraints of CF and EPRL require compactness of land use, while the five-level guidance of UDLS requires scattered and efficient use of land at the same time. In summary, Changchun's territorial spatial planning puts forward high requirements for the efficient use of land and constraints on red line areas. Our research provides a good interpretation of these requirements in terms of spatial layout.

According to the simulation results in Section 3.3, the protection of EPRL and CF policy would affect the future land-use distribution. In order to protect the resources of arable land, woodland, and grassland, these two policies should be strictly observed when implementing territorial spatial policies. In the future territorial spatial planning of Changchun, decision-makers should fully consider the impact of ecological space constraints on future land expansion, and strictly control the reduction in ecological functional areas. Furthermore, affected by the results of UDLS, built-up land would show a trend of

decentralized growth in the future, indicating the importance of limiting the increase of urban land and improving the economical use of land. In addition, it is also necessary to fully explore the existing land resources in Changchun.

## 5. Conclusions

An improved land-use simulation model with dynamically nested ecological spatial constraints (LSDNE) was developed on the basis of EPRL, CF, and UDLS. Driving factors of land expansion were analyzed by RF, and future land-use distribution was simulated using an MRPS-based CA model. We applied our method to Changchun, China. The improved method coupled multilevel ecological spatial constraints as an adjustment module of the LSDNE model and showed the driving factors behind each type of land expansion and the strength of their impact intensity, with high simulation accuracy. From 2010 to 2020, the arable land of Changchun consistently became more extensive and scattered. Due to the occupation of arable land, Changchun had the largest increase in built-up land. Terrain elevation was the most significant factor in all kinds of land expansion. In terms of human and economic factors, population and proximity to highways and railways also played significant roles in built-up land expansion. Five scenarios were designed to evaluate the impacts of the different ecological spatial constraints on future land expansion. Under S1 (inertial development) and S2 (EPRL), the patches were smaller and more fragmented compared with other scenarios. Furthermore, the connectivity between the patches was better under S3 (CF), S4 (EPRL and CF), and S5 (UDLS, EPRL, and CF). Overall, this study provides accurate support for the efficient use of land resources and future spatial planning, and it is of significance for researching other cities or scales. However, the limitations of the data acquisition may prevent some driving factors from being considered, likely affecting the accuracy of simulation. Uncertainty may arise from the quantitative simulation using the Markov model, and further research is needed in the future.

**Author Contributions:** Conceptualization, C.L.; methodology, C.L.; software, C.L.; validation, C.L.; formal analysis, C.L.; investigation, C.L., J.S. and S.S.; data curation, C.L.; writing—original draft preparation, C.L.; writing—review and editing, C.L. and R.L.; visualization, C.L.; supervision, R.L. and Z.S.; project administration, R.L.; funding acquisition, R.L. All authors have read and agreed to the published version of the manuscript.

**Funding:** This research was funded by the National Natural Science Foundation of China, grant number 52170186.

**Data Availability Statement:** The data that support the findings of this study are available from the corresponding author upon reasonable request.

**Acknowledgments:** The authors thank the Changchun Institute of Urban Planning and Design, as well as the Ecology and Environment Bureau of Changchun, for their assistance in data preparation. Furthermore, the authors appreciate the editors and reviewers for their constructive comments and suggestions.

**Conflicts of Interest:** The authors declare no conflict of interest.

## Abbreviations

| Abbreviations | Full Name |
| --- | --- |
| CA | Cellular automata |
| RF | Random forest |
| TAS | Transition analysis strategy |
| PAS | Pattern analysis strategy |
| LSDNE | Land-use simulation model with dynamically nested ecological spatial constraints |
| MRPS | Multitype random patch seeds |
| UDLS | Urban development land-use suitability |
| CF | Capital farmland |
| EPRL | Ecological protection red line |

## Appendix A

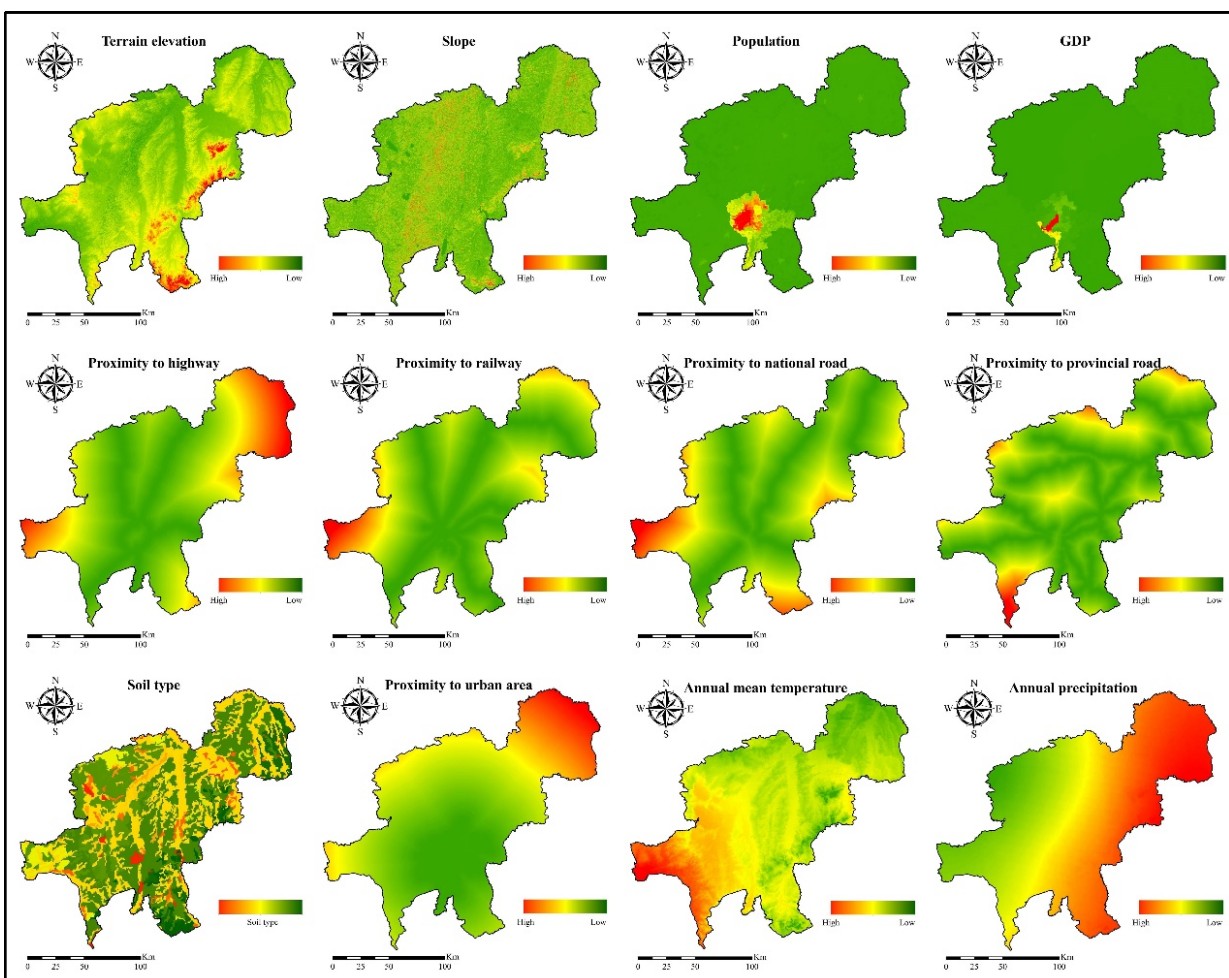

**Figure A1.** Maps of driving factors.

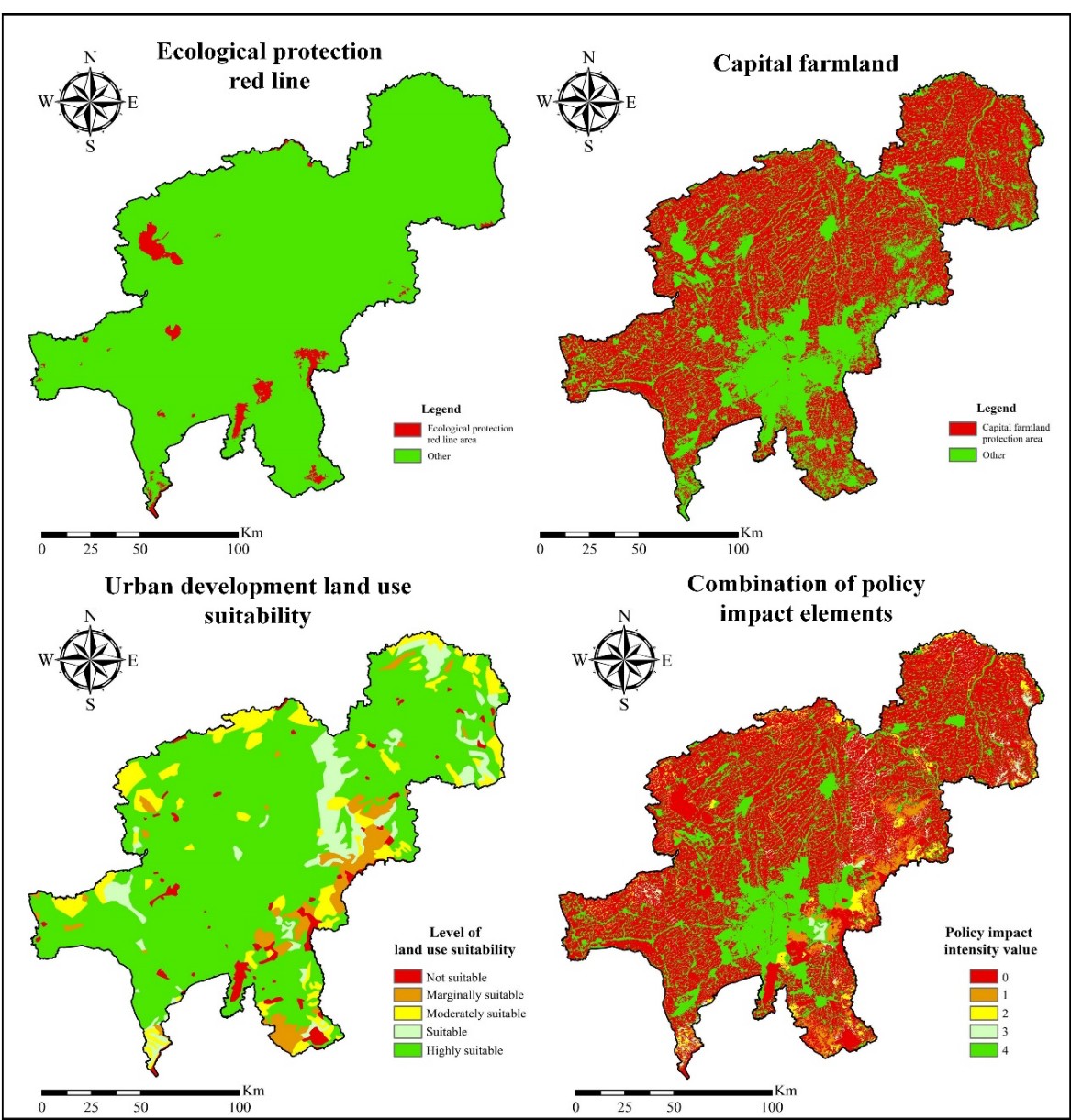

**Figure A2.** Maps of ecological spatial constraints.

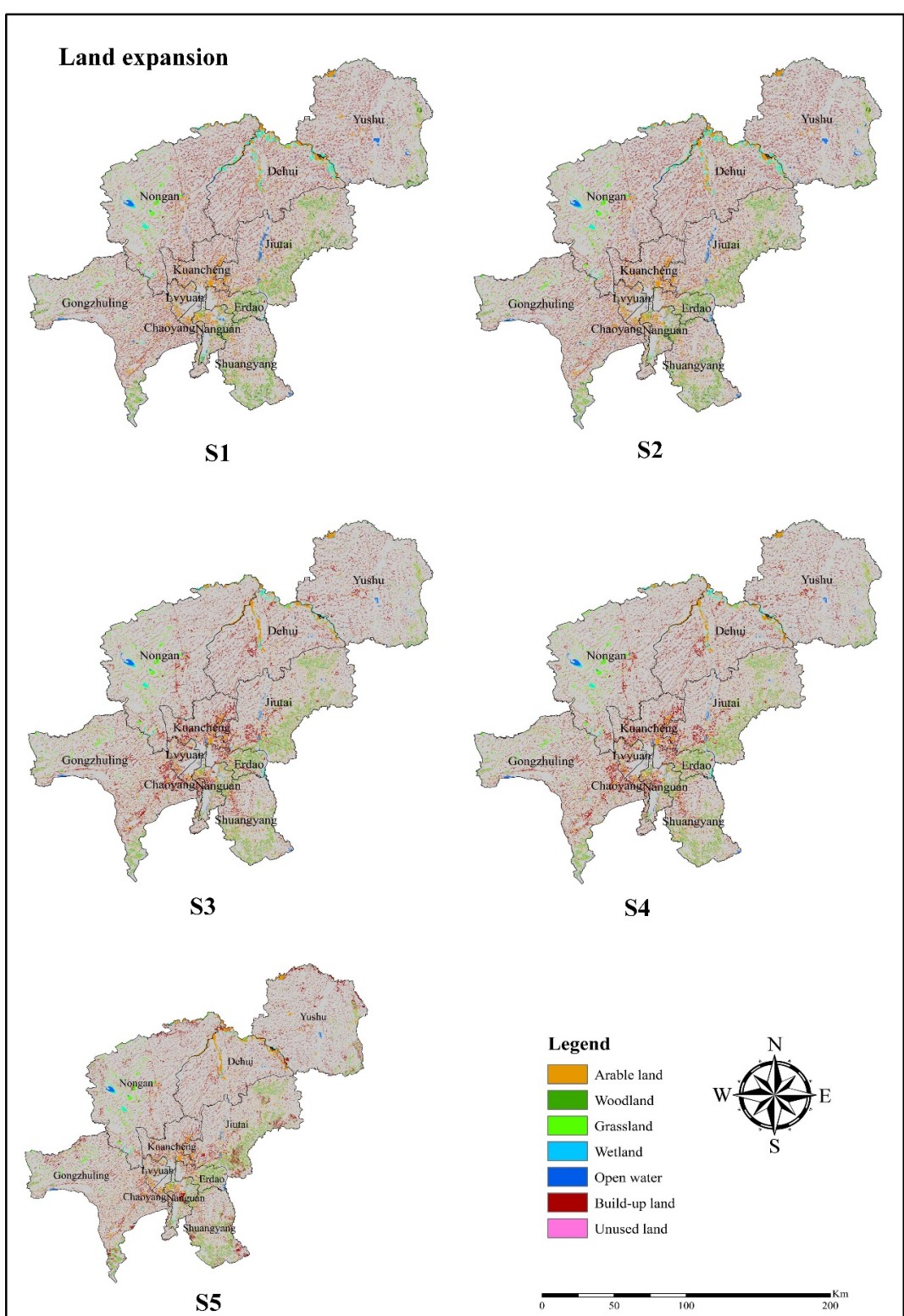

**Figure A3.** Land expansion maps in 2030 under five scenarios.

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
