# Peer review of "An Improved Future Land-Use Simulation Model with Dynamically Nested Ecological Spatial Constraints"

_remotesensing, doi:10.3390/rs15112921_

Round 1

Reviewer 1 Report

It is interesting to develop a called LSDNE land use simulation model which combining the dynamic multi-level ecological spatial contraints into CA model. Moreover, the multi-type random patch seeds were used to analyze the driving factors behind each type of land expansion. This LSDNE mode were used to predict the future land change with five different scenarios. Generally, this work is interesting and is helpful for the Territorial Spatial Planning Policy implementation in China. Some comments  are listed below.

1. The Land Use Simulation model with Dynamic Nested Ecological spatial constraints, the model of LSDNE. It is not very clearly for readers to understand what is "dynamic nested".  The authors should give more detail description about dynamic nested.

2. In section 2, the method of evalution or accuracy assesment or sensitivity analysis for the developed simulation model should be described.

3.  2.3.2 Landscape metrics calculation, I don't know why those landscape metrics should be caculated, for what purposes? I also didn't find any explainations in the Introduction section.

4. Again, without any accuracy assesment or sensitivity analysis results it is hard to compare the differences among the five different scenarios and which one is better.

Reviewer 2 Report

See PDF document with my report.

Round 2

Reviewer 2 Report

The authors have satisfactorily modified the original submission. In my opinion, the manuscript can now be considered for publication in this jounal.